

# Nitrous oxide (N₂O) and methane (CH₄) in rivers and estuaries of northwestern Borneo

Hermann W. Bange[1], Chun Hock Sim[2], Daniel Bastian[1], Jennifer Kallert[1], Annette Kock[1], Aazani Mujahid[3] and Moritz Müller[2]

[1] GEOMAR Helmholtz Centre for Ocean Research Kiel, Kiel, Germany

[2] Swinburne University of Technology, Faculty of Engineering, Computing and Science, Kuching, Sarawak, Malaysia

[3] Department of Aquatic Science, Faculty of Resource Science & Technology, University Malaysia Sarawak, Kota Samarahan, Sarawak, Malaysia

Correspondence to: Hermann Bange, hbange@geomar.de

ms in preparation for *Biogeosciences - Special Issue 'Biogeochemical processes in highly dynamic peat-draining rivers and estuaries in Borneo'*

*ORCID# (https://orcid.org/)*
*HWB: 0000-0003-4053-1394*
*CHS: not available*
*DB: 0000-0002-5102-7399*
*JK: not available*
*AK: 0000-0002-1017-605*
*AM: not available*
*MM: 0000-0001-8485-1598*

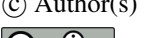



Abstract

Nitrous oxide ($N_2O$) and methane ($CH_4$) are atmospheric trace gases which play important roles of the climate and atmospheric chemistry of the Earth. However, little is known about their emissions from rivers and estuaries which seem to contribute significantly to the atmospheric budget of both gases. To

this end concentrations of $N_2O$ and $CH_4$ were measured in the Rajang, Maludam, Sebuyau and Simunjan Rivers draining peatland in northwestern (NW) Borneo during two campaigns in March and September 2017. The Rajang River was additionally sampled in August 2016 and the Samusam and Sematan Rivers were additionally sampled in March 2017. The Maludam, Sebuyau, and Simunjan Rivers are typical 'blackwater' rivers with very low pH, very high dissolved organic carbon (DOC)

concentrations and very low $O_2$ concentrations. The spatial and temporal variability of $N_2O$ and $CH_4$ concentrations (saturations) in the six rivers/estuaries was large and ranged from 2.0 nmol $L^{-1}$ (28 %) to 41.4 nmol $L^{-1}$ (570 %) and from 2.5 nmol $L^{-1}$ (106 %) to 1372 nmol $L^{-1}$ (57,459 %), respectively. We found no overall trends of $N_2O$ with $O_2$ or $NO_3^-$, $NO_2^-$, $NH_4^+$ and there were no trends of $CH_4$ with $O_2$ or dissolved nutrients or DOC. $N_2O$ concentrations showed a positive linear correlation with

rainfall. We conclude, therefore, that rainfall is the main factor determining the riverine $N_2O$ concentrations since $N_2O$ production/consumption in the 'blackwater' rivers themselves seems to be unlikely because of the low pH. In contrast $CH_4$ concentrations showed an inverse relationship with rainfall. $CH_4$ concentrations were highest at salinity = 0 and most probably result from methanogenesis as part of the decomposition of organic matter under anoxic conditions. We speculate

that $CH_4$ oxidation, which can be high when the water discharge is high (e.g. after rainfall events), is responsible for the decrease of the $CH_4$ concentrations along the salinity gradients. The rivers and estuaries studied here were an overall net source of $N_2O$ and $CH_4$ to the atmosphere. The total annual $N_2O$ and $CH_4$ emissions were 1.09 Gg $N_2O$ $yr^{-1}$ (0.7 Gg N $yr^{-1}$) and 23.8 Gg $CH_4$ $yr^{-1}$, respectively. This represents about 0.3 – 0.7 % of the global annual riverine and estuarine $N_2O$ emissions and about

0.1 – 1 % of the global riverine and estuarine $CH_4$ emissions. Therefore, we conclude that rivers and estuaries in NW Borneo –despite the fact their water area covers only 0.05 % of the global river/estuarine area– contribute significantly to global riverine and estuarine emissions of $N_2O$ and $CH_4$.






## 1. Introduction

Nitrous oxide ($N_2O$) and methane ($CH_4$) are atmospheric trace gases which influence the climate and atmospheric chemistry of the Earth (IPCC, 2013; WMO, 2014). They act as greenhouse gases in the

troposphere and are indirectly involved in stratospheric ozone depletion. Emission estimates indicate that rivers and estuaries contribute significantly to the atmospheric budget of both $N_2O$ and $CH_4$. $N_2O$ emissions estimate for rivers and estuaries range from 0.05 to 3.3 Tg $N_2O$ $yr^{-1}$ and from 0.09 to 5.7 Tg $N_2O$ $yr^{-1}$, respectively (see overview in (Maavara et al., 2019). Thus, the combined riverine and estuarine emissions may contribute up to 32 % to the global natural and anthropogenic emissions of

$N_2O$ (28.1 Tg $N_2O$ $yr^{-1}$; IPCC, 2013). $CH_4$ emission estimates for rivers and estuaries are in the range of 1.5 – 26.8 Tg $CH_4$ $yr^{-1}$ (Bastviken et al., 2011; Stanley et al., 2016) and 0.8 – 6.6 Tg $CH_4$ $yr^{-1}$ (see overview in (Borges and Abril, 2011)), respectively. The combined emissions from rivers and estuaries can contribute up to 6% of the global natural and anthropogenic atmospheric emissions of $CH_4$ (556 Tg $CH_4$ $yr^{-1}$; (IPCC, 2013)). As indicated by the wide range of the estimates cited above, the

emission estimates of both gases are associated with a high degree of uncertainty, which is mainly caused by an inadequate coverage of the temporal and spatial distributions of $N_2O$ and $CH_4$ in rivers and estuaries and the inherent errors of the model approaches to estimate their release across the water/atmosphere interface (see e.g. (Alin et al., 2011; Borges and Abril, 2011)).

$N_2O$ is produced by microbial processes such as nitrification (i.e. oxidation of ammonia, $NH_3$, to nitrite, $NO_2^-$) in estuarine waters (see e.g. (Barnes and Upstill-Goddard, 2011)) and heterotrophic denitrification (i.e. reduction of nitrate, $NO_3^-$, to dinitrogen, $N_2$) in river sediments (Beaulieu et al., 2011). The yields of $N_2O$ from these processes are enhanced under low oxygen (i.e. suboxic) conditions (see e.g. (Brase et al., 2017; Zhang et al., 2010)), whereas $N_2O$ can be reduced to $N_2$ under

anoxic conditions via sedimentary denitrification in rivers (see e.g. (Upstill-Goddard et al., 2017)). Apart from ambient oxygen ($O_2$) concentrations, riverine and estuarine $N_2O$ production is also dependent on the concentrations of dissolved inorganic nitrogen, DIN ($= NH_4^+ + NO_{2-} + NO_3^-$). There seems to be a general trend towards high estuarine $N_2O$ concentrations when DIN concentrations are high as well (Barnes and Upstill-Goddard, 2011; Zhang et al., 2010). However, this trend masks the

fact that in many cases the spatial and temporal variability of riverine and estuarine $N_2O$ is often not related to DIN (see e.g. (Borges et al., 2015; Brase et al., 2017; Müller et al., 2016a)).

$CH_4$ is produced during microbial respiration of organic matter by anaerobic methanogenesis in riverine and estuarine sediments (see e.g. (Borges and Abril, 2011; Romeijn et al., 2019; Stanley et al.,

2016)). A significant fraction of the $CH_4$ produced in sediments can be oxidized to carbon dioxide ($CO_2$) via anaerobic $CH_4$ oxidation in sulphate-reducing zones of estuarine sediments (see e.g. (Maltby et al., 2018)). When released to the overlying riverine/estuarine water $CH_4$ can be oxidized by aerobic





CH$_4$ oxidation before reaching the atmosphere (see e.g. (Borges and Abril, 2011; Sawakuchi et al., 2016; Steinle et al., 2017)).


In general, the temporal and spatial distributions of N$_2$O and CH$_4$ in rivers and estuaries are driven by the complex interplay of microbial production and consumption pathways (see above) as well as physical processes such as input via shallow groundwater, river discharge, tidal pumping, release to the atmosphere and export to coastal waters (Barnes and Upstill-Goddard, 2011; Borges and Abril, 105    2011; Stanley et al., 2016).

Peatlands, which are found in the tropics and at high latitudes, constitute one of the largest reservoirs of organic-bound carbon worldwide (Page et al., 2011; Treat et al., 2019; Yu et al., 2010). Rivers and streams draining peatlands have exceptionally high concentrations of dissolved organic carbon (DOC) and low pH and, thus, belong to the 'blackwater' river type which is also found in southeast (SE) Asia 110    (Alkhatib et al., 2007; Baum et al., 2007; Martin et al., 2018; Moore et al., 2011; Rixen et al., 2008; Wit et al., 2015)..

Despite the fact that a number of studies about N$_2$O and CH$_4$ emissions from peatlands in southeast 115    (SE) Asia have been published (see e.g. (Couwenberg et al., 2010; Hatano et al., 2016; Jauhiainen et al., 2012), only a few studies about their emissions from peatland draining rivers in SE Asia have been published so far (Jauhiainen and Silvennoinen, 2012; Müller et al., 2016a). Therefore, our knowledge about the biogeochemistry and emissions of N$_2$O and CH$_4$ from peatland draining rivers is still rudimentary at best.


Here we present measurements of dissolved N$_2$O and CH$_4$ in six rivers and their estuaries in northwestern (NW) Borneo during August 2016, March 2017 and September 2017. The objectives of our study were (i) to measure the distributions of dissolved N$_2$O and CH$_4$, (ii) to identify the major factors influencing their distributions and (iii) to estimate the N$_2$O and CH$_4$ emissions to the 125    atmosphere.

## 2. Study site description

Discrete samples of surface water were taken at several stations along the salinity gradients of the 130    Rajang, Maludam, Sebuyau and Simunjan Rivers in NW Borneo during two campaigns in March and September 2017 (Figure 1, Table 1). The Rajang River was additionally sampled in August 2016 and the Samusam and Sematan Rivers were additionally sampled in March 2017. The environmental settings of the river basins are summarized in Table 2. Based on the areas affected by oil palm plantations and logging in combination with our own observations during several samplings



campaigns, we classified the Rajang and Simunjan river basins as 'disturbed', the Maludam, Sebuyau,
Sematan and Samusam river basins as 'undisturbed' (Table 2).

### 3. Methods


3.1 Measurements of $N_2O$ and $CH_4$

Discrete water samples were taken as duplicates or triplicates in 20 or 37 mL glass vials from a water
depth of approximately 1 m. The samples were poisoned immediately after sampling with a saturated
aqueous mercuric chloride ($HgCl_2$) solution. The samples were shipped to GEOMAR Helmholtz

Centre for Ocean Research Kiel, Germany, for further analysis within a few weeks after sampling. For
the determination of the $N_2O$ and $CH_4$ concentrations we applied the static-headspace equilibration
method followed by gas chromatographic separation and detection with an electron capture detector
(ECD, for $N_2O$) and a flame ionization detector (FID, for $CH_4$) as described in (Bastian, 2017) and
(Kallert, 2017). Calibration of the ECD and FID were performed with standard gas mixtures of $N_2O$

and $CH_4$ in synthetic air which have been calibrated against NOAA-certified primary gas standards.

Dissolved $N_2O/CH_4$ concentrations ($C_{obs}$ in nmol $L^{-1}$) were calculated with

$$C_{obs} = x'PV_{hs}/(RTV_{wp}) + x'\beta P \qquad\qquad (1),$$


where $x'$ is the dry mole fraction of $N_2O$ or $CH_4$ in the headspace of the sample, $P$ is the ambient
pressure (set to 1013.25 hPa), $V_{hs}$ and $V_{wp}$ are the volumes of the headspace and the water phase,
respectively. $R$ stands for the gas constant (8.31451 $m^3$ Pa $K^{-1}$ $mol^{-1}$), $T$ is the temperature during
equilibration and $\beta$ is the solubility of $N_2O$ or $CH_4$ (Weiss and Price, 1980; Wiesenburg and Guinasso

Jr., 1979). The estimated mean relative errors of the measurements were +/- 9 % and +/- 13 % for $N_2O$
and $CH_4$, respectively. These comparably high relative errors most probably resulted from the long
storage time for some of the samples. It was shown that $CH_4$ samples are more sensitive to storage
time than $N_2O$ samples (Wilson et al., 2018).

3.2 Ancillary measurements

Water temperature, dissolved oxygen, and salinity were recorded with an Aquaread® 2000. Nutrient
measurements are described in detail in (Sia et al., 2019). In short, all samples were collected within
the upper 1 m (surface) using pre-washed bottles via a pole-sampler to reduce contamination from the
surface of the boat and engine coolant waters (Zhang et al., 2015). Samples were filtered through a 0.4

μm pore-size polycarbonate membrane filters (Whatman) into pre-rinsed bottles, killed with
concentrated $HgCl_2$ solution and kept in a cool, dark room. Nutrients were determined utilizing a



Skalar SANplus auto analyser with an analytical precision <5%. The measurements of dissolved organic carbon (DOC) are described in detail in (Martin et al., 2018). The DOC data are available from the supplementary material in (Martin et al., 2018).


### 3.3 Computations of saturations and flux densities

The saturations (*Sat*, %) for $N_2O$, $CH_4$ and $O_2$ were calculated as

$$Sat = 100 \, C_{obs} \, / \, C_{eq} \qquad (2)$$


where $C_{eq}$ is the equilibrium concentration of $N_2O$/$CH_4$/$O_2$ calculated according to (Weiss and Price, 1980), (Wiesenburg and Guinasso Jr., 1979) or (Weiss, 1970), respectively, with the *in-situ* temperature and salinity as well as the mean dry mole fractions of $N_2O$/$CH_4$ at the time of the sampling. Mean monthly $N_2O$/$CH_4$ dry mole fractions of 329/1841 $10^{-9}$ (ppb), 331/1880 ppb and

330/1852 ppb for August 2016, March 2017 and September 2017, respectively, were measured at the atmospheric monitoring station Bukit Kototabang, located on the west coast of Sumatra (Indonesia). This station is operated by the NOAA/ESRL Global Monitoring Division program and data are available from http://www.esrl.noaa.gov/gmd. A saturation < 100 % indicates a concentration lower than the theoretical equilibrium concentration (i.e. undersaturation) and a saturation > 100 % indicates

supersaturation.

Flux densities (*F*, nmol·m$^{-2}$·s$^{-1}$) were calculated as

$$F = k_w \, (C_{obs} - C_{eq}) \qquad (3)$$
$$k_w = k_{600} \, (Sc/600)^{-0.5} \qquad (4)$$

$k_w$ is the gas transfer velocity and $Sc$ is the Schmidt number, which was calculated with the equations for the kinematic viscosity of water (Siedler and Peters, 1986) and the diffusion of $N_2O$ or $CH_4$ in water (Jähne et al., 1987; Rhee et al., 2009). $k_{600}$ was determined in a seasonal study for the Lupur and

Saribas Rivers which are located in close vicinity to the Maludam River (Müller et al., 2016a; Müller et al., 2016b). We assume that the $k_{600}$ values measured by (Müller et al., 2016a)are representative for the rivers in NW Borneo studied here. Mean $k_{600}$ range from 13.2 cm h$^{-1}$ (Lupur River) to 23.9 cm h$^{-1}$ (Saribas River tributary). On the basis of the data in (Müller et al., 2016a) we computed a mean $k_{600}$ of 19.2 cm h$^{-1}$ (5.33 $10^{-5}$ m s$^{-1}$) which we used to estimate the flux densities of $N_2O$ and $CH_4$. This $k_{600}$ is

in good agreement with the mean $k_{600}$ for rivers and estuaries listed in (Alin et al., 2011) which range from 4.8 to 35.3 cm h$^{-1}$.

### 3.4 Rainfall data



In order to account for the regional variability of the rainfall in NW Borneo, we used mean monthly
rainfall data recorded at the weather stations in Kuching, Bandar Sri Aman and Sibu (all in NW
Borneo). The rainfall data were provided by World Weather Online (Dubai, UAE, and Manchester,
UK) and are available via https://www.worldweatheronline.com/. Representative weather stations
were chosen for each river basin studied here and allocated as follows: The rainfall data for the
Simunjan, Sematan and Samsuman River basins are represented by the data from Kuching, the
Maludam/Sebuyau and the Rajang River basins are represented by the data from the Bandar Sri Aman
and Sibu weather stations, respectively.

## 4    Results and Discussion

All rivers showed low concentrations of DIN in the range from 1.1 to 29 µmol L$^{-1}$ (Table 1). NO$_3^-$
concentrations ranged from below the detection limit of 0.14 µmol L$^{-1}$ up to 19 µmol L$^{-1}$ and NH$_4^+$
concentrations were in the range of 0.3 to 17 µmol L$^{-1}$. The Maludam, Sebuyau, and Simunjan Rivers
can be classified as 'blackwater' rivers with low pH (3.7 – 4.8), high DOC concentrations (1960 –
4387 µmol L$^{-1}$) and low O$_2$ concentrations (31 – 95 µmol L$^{-1}$; 13 – 39 % saturation) at salinity = 0
(Table 1). Comparable settings have been reported from other tropical 'blackwater' rivers in SE Asia
as well (Alkhatib et al., 2007; Baum et al., 2007; Moore et al., 2011; Rixen et al., 2008; Wit et al.,
2015).

### 4.1 Nitrous oxide

The measured ranges of N$_2$O concentrations and saturations are listed in Table 3 and the distributions
of N$_2$O saturations along the salinity gradients are shown in Figure 2. N$_2$O concentrations (saturations)
were highly variable and ranged from 2.0 nmol L$^{-1}$ (28 %) in the Rajang River (at salinity = 0 in
August 2016) to 41.4 nmol L$^{-1}$ (570 %) in the Simunjan River (at salinity = 0 in March 2017). N$_2$O
concentrations in the Rajang, Maludam and Sebuyau Rivers were generally higher in September 2017
compared to March 2017 (Figure 2a-c). A decreasing linear trend of the N$_2$O saturations with salinity
was only observed for the Rajang River in March 2017 (Figure 2a) indicating a conservative mixing
and no N$_2$O sources or sinks along the salinity gradient. Our results are in general agreement with the
N$_2$O measurements in the Lupar and Saribas Rivers (which are located in close vicinity of the
Maludam River) in June 2013 and March 2014: Müller et al. (2016) measured N$_2$O concentrations
(saturations) from 6.6 to 117 nmol L$^{-1}$ (102 to 1679 %) in the Lupar and Saribas Rivers. Salinity and
N$_2$O concentrations in the Lupar and Saribas Rivers were negatively correlated in June 2013 but were
not correlated in March 2014 (Müller et al., 2016a). In contrast to our study, no N$_2$O undersaturations
have been observed by (Müller et al., 2016a). Our results are at the lower end of N$_2$O concentrations
reported from rivers around the globe which can range from extreme undersaturation (down to about 3
%, i.e. almost devoid of N$_2$O) as measured in a tropical river in Africa (Borges et al., 2015; Upstill-



Goddard et al., 2017) to extreme supersaturation (of up to 6500%) as measured in a river in Europe ((Barnes and Upstill-Goddard, 2011).

Maximum $N_2O$ saturations measured in March 2017 were in the range from 106 % to 142 % for the
rivers classified as undisturbed (Maludam, Sebuyau, Sematan and Samusam) whereas the maximum saturation for the rivers classified as disturbed (Rajang and Simunjan) were in the range from 329 % to 570 % (Tables 2 and 3) indicating higher emissions from the disturbed rivers. The maximum $N_2O$ saturations in September 2017 ranged from 329 % to 390 % and no differences were observed between undisturbed and disturbed rivers (Table 3).


We found no overall trends of $N_2O$ with $O_2$ or $NO_3^-$, $NO_2^-$, $NH_4^+$ and DIN. Therefore, it is difficult to decipher the major consumption or production processes of $N_2O$ or to locate the influence of (local) anthropogenic input of nitrogen compounds on riverine $N_2O$ cycling. This is in line with results from studies of other tropical rivers (Borges et al., 2015; Müller et al., 2016a). $N_2O$ production via
nitrification depends on the prevailing pH because nitrifiers prefer to take up ammonia ($NH_3$). The concentration of dissolved $NH_3$ is dropping significantly at pH < 8-9 (Bange, 2008) because of its easy protonation to ammonium ($NH_4^+$). A low pH of about 5-6 can reduce nitrification ($NH_4^+$ oxidation) significantly as it was recently shown for the Tay Ninh River in Vietnam (Le et al., 2019). Moreover, the optimum for a net $N_2O$ production by nitrification, nitrifier-denitrification and denitrification lies
between a pH of 7 – 7.5 (Blum et al., 2018). Therefore, a net $N_2O$ production may be unlikely in the 'blackwater' rivers studied here with their low pH (see Table 1). The observed $N_2O$ supersaturations, therefore, might have been the result of external inputs of $N_2O$-enriched waters or groundwater. The observed $N_2O$ undersaturations were most probably resulting from heterotrophic denitrification which could have taken place either in organic matter-enriched anoxic river sediments or in anoxic
environments of the surrounding soils. However, the main factor for riverine $N_2O$ under- or supersaturation might be rainfall, because rainfall events determine the height of the water table in the surrounding soils, which in turn determines the amount of suboxic/anoxic conditions favourable for $N_2O$ production or consumption. See also discussion in Section 4.3.

4.2 Methane
The measured ranges of $CH_4$ concentrations and saturations are listed in Table 3 and the distributions of $CH_4$ saturations along the salinity gradients are shown in Figure 3. $CH_4$ concentrations (saturations) were highly variable and ranged from 2.5 nmol $L^{-1}$ (106 %) in the Simunjan River (at salinity = 0 in September 2017) to 1372 nmol $L^{-1}$ (57,459 %) in the Simunjan River (at salinity = 0 in March 2017).
(Please note that we also measured a $CH_4$ concentration of 14,999 nmol $L^{-1}$ (624,070 %) at one station in the Simunjan River at salinity = 0 in March 2017 which, however, was not included in Figure 3 and which was not used in further computations because of statistical reasons.) $CH_4$ saturations in the





Rajang, Maludam, Sebuyau and Simunjan Rivers were higher in March 2017 compared to September

2017. Maximum $CH_4$ concentrations were measured at salinity = 0 and there was a general decrease of

$CH_4$ with increasing salinity. Exceptions from this trend occurred at individual stations in the

Maludam, Sebuyau and Samusam Rivers which point to local sources of $CH_4$ (Figure 3). The range of

$CH_4$ concentrations (saturations) from our study is larger compared to the concentration range

measured in the Lupar and Saribas Rivers (3.7 – 113.9 nmol L$^{-1}$; 168 – 5058 %) ((Müller et al.,

2016a). (Borges et al., 2015) reported a maximum $CH_4$ concentration (saturation) of 62,966 nmol L$^{-1}$

(appr. 954,000 %) in their study of tropical rivers in Africa which is much higher than the maximum

concentration measured in our study. (Bouillon et al., 2014)

We found no overall trends of $CH_4$ with $O_2$ or dissolved nutrients or DOC along the salinity gradients.

High $CH_4$ concentrations, which were often associated with high DOC and low $O_2$ concentrations at

salinity = 0, might have been produced by methanogenesis in anoxic riverine sediments rich in organic

material or in anoxic parts of the surrounding soils drained by the rivers. The decrease of $CH_4$ with

increasing salinity can be attributed to the gas exchange across the river water/atmosphere interface in

combination with $CH_4$ oxidation (Borges and Abril, 2011; Sawakuchi et al., 2016).

We found no differences in the $CH_4$ saturations between the rivers classified as undisturbed and those

classified as disturbed in both March and September 2017.

### 4.3 $N_2O$/$CH_4$ concentrations and rainfall

Mean $N_2O$ concentrations showed a linear correlation with rain fall (Figure 4a). Enhanced $N_2O$

emissions from (peat) soils are usually associated with rainfall when the water table approaches the

soil surface (Couwenberg et al., 2010; Jauhiainen et al., 2016). A high water table, in turn, allows

decomposition of previously deposited fresh organic material (Jauhiainen et al., 2016) and, thus, will

result in favourable conditions for microbial $N_2O$ production mainly via denitrification in a

suboxic/anoxic soil environment (Pihlatie et al., 2004). $N_2O$ production via nitrification may be less

important at high water table (Pihlatie et al., 2004). Therefore, the positive linear relationship of the

riverine $N_2O$ concentrations with rainfall might result from enhanced $N_2O$ production in the adjacent

soils drained by the rivers.

In contrast to $N_2O$, the mean $CH_4$ concentrations decrease with increasing rainfall (Figure 4b). Under

the assumption that rainfall is a predictor for river discharge/high water we can argue that our result

are in agreement with (i) the often observed inverse relationship between $CH_4$ concentrations and river

discharge (Anthony et al., 2012; Bouillon et al., 2014; Dinsmore et al., 2013; Hope et al., 2001) and

(ii) the enhancement of $CH_4$ oxidation during high waters: (Sawakuchi et al., 2016) showed that $CH_4$

oxidation in 'blackwater' rivers of the Amazon basin was maximal during the high water season





resulting in a reduction of up to 96% of the diffusive flux of $CH_4$ (i.e. its input to the river and its

release to the atmosphere) (Sawakuchi et al., 2016). This was explained by the higher river water

levels which, in turn, could enhance $CH_4$ oxidation because of a longer residence time of $CH_4$ in the

sediment and river water (Sawakuchi et al., 2016).

### 4.4 Emission estimates

The $N_2O$ flux densities from the six rivers studied here are comparable to the $N_2O$ flux densities from

other aqueous and soil systems reported from Borneo and other sites in SE Asia, see Table 4. The

corresponding $CH_4$ flux densities are higher than the $CH_4$ flux densities reported for the Lupar and

Saribas Rivers but much lower than the flux densities from drainage canals in Central Kalimantan and

Sumatra (Jauhiainen and Silvennoinen, 2012) (Table 4). Our $CH_4$ flux densities are, however,

comparable to recently published $CH_4$ eddy covariance measurements (Tang et al., 2018) in the

Maludam National Park, which is drained by the Maludam River, and measurements of the $CH_4$

release from peat soils when the water table is high and $CH_4$ from rice paddies (Couwenberg et al.,

2010), see Table 4. The mean annual $N_2O$ and $CH_4$ emissions for the individual rivers were calculated

by multiplying the mean flux density, $F$, for each river (Table 4) with the river surface area given in

Table 2. The results are listed in Table 5. The resulting total annual $N_2O$ emissions for the rivers in

NW Borneo -including the emissions from the Lupar and Saribas Rivers (Müller et al., 2016a)- are

1.09 Gg $N_2O$ yr$^{-1}$ (0.7 Gg N yr$^{-1}$). This represents about 0.3 – 0.7 % of the global annual riverine and

estuarine $N_2O$ emissions of 166 – 322 Gg $N_2O$ (106 – 205 Gg N yr$^{-1}$) recently estimated by (Maavara

et al., 2019). The total annual $CH_4$ emissions from rivers in NW Borneo are 23.8 Gg $CH_4$ yr$^{-1}$. This

represents about 0.1 – 1 % of the global riverine and estuarine $CH_4$ emissions of 2300 – 33,400 Gg

$CH_4$ yr$^{-1}$ (the emission range is based on the minimum and maximum estimates given in (Bange et al.,

1994; Bastviken et al., 2011; Borges and Abril, 2011; Stanley et al., 2016). However, we caution that

our estimates are associated with a high degree of uncertainty because (i) our data are biased by the

fact that for some rivers it was not possible to cover the entire salinity gradient and (ii) seasonal and

internannual variabilities are not adequately represented in our data set.

### 5    Summary and Conclusions

$N_2O$ and $CH_4$ were measured in the Rajang, Maludam, Sebuyau and Simuntan Rivers and Estuaries in

NW Borneo during two campaigns in March and September 2017. The Rajang River was additionally

sampled in August 2016 and the Samusam and Sematan Rivers were additionally sampled in March

2017. The spatial and temporal variability of $N_2O$ and $CH_4$ concentrations was large. $N_2O$

concentrations (saturations) ranged from 2.0 nmol L$^{-1}$ (28 %) in the Rajang River (at salinity = 0 in

August 2016) to 41.4 nmol L$^{-1}$ (570 %) in the Simunjan River (at salinity = 0 in March 2017). $CH_4$

concentrations (saturations) were in the range from 2.5 nmol L$^{-1}$ (106 %) in the Simunjan River (at





salinity = 0 in September 2017) to 1372 nmol L$^{-1}$ (57,459 %) in the Simunjan River (at salinity = 0 in
March 2017). N$_2$O concentrations showed a positive linear correlation with rainfall. We conclude,
therefore, that rainfall, which determines the N$_2$O production/consumption in the surrounding soils, is

the main factor determining the riverine N$_2$O concentrations. N$_2$O production in the 'blackwater'
rivers themselves seems to be unlikely because of the low pH. In contrast CH$_4$ concentrations showed
an inverse relationship with rainfall. CH$_4$ concentrations were highest at salinity = 0 and most
probably results from methanogenesis as part of the decomposition of organic matter under anoxic
conditions. We speculate that CH$_4$ oxidation, which can be high when the water discharge is high (e.g.

after rainfall events), is responsible for the reduction of the CH$_4$ concentrations along the salinity
gradient. The rivers and estuaries studied here were an overall net source of N$_2$O and CH$_4$ to the
atmosphere. The total annual N$_2$O and CH$_4$ emissions were 1.09 Gg N$_2$O yr$^{-1}$ (0.7 Gg N yr$^{-1}$) and 23.8
Gg CH$_4$ yr$^{-1}$, respectively. This represents about 0.3 – 0.7 % of the global annual riverine and estuarine
N$_2$O emissions and about 0.1 – 1 % of the global riverine and estuarine CH$_4$ emissions. Rivers and

estuaries in NW Borneo contribute only 0.05 % (= 7.9 10$^2$ km$^2$ including the surface areas of the
Lupar and Saribas Rivers; (Müller et al., 2016a) to the global water surface area of rivers and estuaries
(= 1.7 10$^6$ km$^2$; (Maavara et al., 2019)). Therefore we conclude that rivers and estuaries in NW Borneo
contribute significantly to the global riverine and estuarine emissions of both N$_2$O and CH$_4$.

The environment of Borneo (and SE Asia) is affected by rapid changes due to (i) anthropogenic
activities such as conversion of peatland into oil palm plantations etc. (see e.g. (Austin et al., 2018;
McAlpine et al., 2018; Schoneveld et al., 2019)) and (ii) climatic changes (see e.g. (Sa'adi et al.,
2017a, b; Tang, 2019)) which, in turn, could significantly affect N$_2$O and CH$_4$ emissions from soils
(see e.g. (Jauhiainen et al., 2016; Oktarita et al., 2017)). But little is known about how these changes

will affect N$_2$O and CH$_4$ emissions from aqueous systems such as rivers and estuaries in the future.
The obvious relationship of N$_2$O and CH$_4$ concentrations and rainfall could be used to predict future
concentrations and its associated emissions to the atmosphere. However, the trends of rainfall and
river discharge in Borneo show a high local variability and no general common trend (Sa'adi et al.,
2017a; Tang, 2019). Therefore, predictions of future trends of N$_2$O and CH$_4$ emissions will be

associated with high degree of uncertainty. In order to improve our knowledge to predicted future
changes of N$_2$O and CH$_4$ riverine/estuarine emissions we suggest establishing regular measurements in
the rivers and along the salinity gradients. This will help deciphering the temporal and spatial
variability of N$_2$O and CH$_4$ emissions from tropical rivers and estuaries. Moreover, studies of the
relevant production/consumption pathways (and their main driving factors) for both gases are

required. A suitable framework for this could be the recently published concept of the global N$_2$O
Ocean Observation Network (N2O-ON) (Bange et al., 2019).





## 6  Acknowledgments

We would like to thank the Sarawak Forestry Department and Sarawak Biodiversity Centre for
permission to conduct collaborative research in Sarawak waters under permit numbers
NPW.907.4.4(Jld.14)-161, Park Permit No WL83/2017, and SBC-RA-0097-MM. We are very grateful
to the boatmen who helped us to collect samples, in particular Lukas Chin, Captain Juble, and their
crew during the Rajang River and Eastern Region cruises, and Minhad and Pak Mat while sampling

the Western Region. We are grateful to Claire Evans and Joost Brandsma for their participation in
planning the overall research project and helping to lead expeditions to the Maludam, Sebuyau, and
Simunjan Rivers. Faddrine Yang, Gonzalo Carrasco, Florina Richard, and Fakharuddin Muhamad
assisted greatly during fieldwork and with logistics. We thank Edwin Sia and Faddrine Holt for the
fantastic support of the $N_2O/CH_4$ sampling during the fieldwork campaigns. We acknowledge the help

of Lasse Sieberth with the $N_2O/CH_4$ measurements. All $N_2O$ and $CH_4$ data presented here are available
from the MEMENTO (the MarinE MethanE and NiTrous Oxide) database:
https://memento.geomar.de. M.M. acknowledges funding through Newton-Ungku Omar Fund
(NE/P020283/1), MOHE FRGS 15 Grant (FRGS/1/2015/WAB08/SWIN/02/1) and SKLEC Open
Research Fund (SKLEC-KF201610).

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



8   Tables

Table 1: Overview of sampling and sampled ranges of salinity, pH as well as $O_2$ concentration and
saturation (in %, given in parenthesis) and concentrations of dissolved inorganic nitrogen (DIN = $NO_3^-$
+ $NO_2^-$ + $NH_4^+$), silicate ($SiO_2$) and dissolved organic carbon (DOC). All concentrations are given in
$\mu mol\ L^{-1}$. na stands for not available and Stat. stands for sampling station. DOC data were taken from
(Martin et al., 2018).

| River | Date | # of Stat. | Range of | | | | | |
|---|---|---|---|---|---|---|---|---|
| | | | Salinity | pH | O₂ | DIN | SiO₂ | DOC |
| Rajang | 20 – 27 Aug '16 | 30 | 0 – 32 | 6.5 – 8.1 | 85 – 153 (42 – 73) | 6.7 – 29 | 4.0 – 179 | na |
| | 4 – 7 Mar '17 | 14 | 0 – 30 | 6.0 – 8.2 | 142 – 237 (58- 109) | 8.1 – 18 | 16 – 158 | 96 – 201 |
| | 5 – 14 Sept '17 | 8 | 0 – 18 | 6.9 – 8.2 | 164 – 227 (76 – 90) | 6.7 – 14 | 12 – 98 | na |
| Maludam | 9 Mar '17 | 9 | 0 – 20 | 3.7 – 7.6 | 34 – 213 (13 – 100) | 3.9 – 10 | 5.8 – 32 | 266 – 4387 |
| | 14/15 Sept '17 | 9 | 0 – 15 | 4.1 – 6.7 | 43 – 155 (17 – 74) | 2.1 – 3.0 | 0.1 – 8.0 | 3072 – 3245 |
| Sebuyau | 11 Mar '17 | 11 | 0 – 24 | 4.3 – 7.8 | 43 – 246 (18 – 116) | 2.9 – 13 | 33 – 78 | 206 – 1968 |
| | 15 Sept '17 | 5 | 0 – 10 | 7.2 – 7.7 | 65 – 179 (27 – 75) | 1.1 – 13 | 0.9 – 44 | 235 – 2052 |
| Simunjan | 12 Mar '17 | 6 | 0 – 0.4 | 4.7 – 6.3 | 31 – 81 (13 – 34) | 2.2 – 16 | 73 – 114 | 2016 – 3039 |
| | 17 Sept '17 | 6 | 0 – 4.6 | 4.8 – 6.7 | 95 – 131 (39 – 53) | 2.0 – 13 | 1.4 – 2.6 | 925 – 1960 |
| Sematan | 9 Mar '17 | 5 | 0 – 28 | 6.8 – 8.3 | 184 – 208 (81 – 102) | 5.9 – 10 | 6.3 – 141 | 100 – 240 |
| Samusam | 11 Mar '17 | 5 | 0 – 27 | 6.3 – 8.2 | 174 – 208 (72 – 102) | 3.9 – 6.6 | 9.7 – 98 | 87 – 1188 |






Table 2: Summary of the environmental settings of the river basins. Based on the area percentage of oil palm, logging and our own surveys and observations, we classified the river basins into undisturbed (U) and disturbed (D). All areas are given in km².

| River | Areas | | | | River water surface⁴ | Remarks | Classification |
| | Total Basin | Peatland¹ | Oil palm plantations² | Logging³ | | | |
|---|---|---|---|---|---|---|---|
| Rajang | 50,000[5] | 3844 | 4514 | 29,379 | 455[5] | The longest river in Malaysia. Major town is Sibu (163,000 population). Smaller townships are Kapit, Kanowit and Sarikei. There is a large number of villages and longhouses (traditional buildings inhabited by local communities) located along the river and its tributaries. Two hydroelectric power plants were built at two tributaries in the upper Rajang basin. The river mouth is surrounded by peat lands, and most of these peat lands have been converted to commercial oil palm plantations. | D |
| Maludam | 197 | 172 | 16 | 0 | 0.36 | The upstream of the river is surrounded by the Maludam National Park. The Maludam Peninsula is bordered by the Lupar and Saribas Rivers and is the biggest undisturbed peat forest in Malaysia. The National Park had been subjected to selective logging before it was gazetted as a totally protected area in 2000. Well preserved peat land. There are oil palm cultivations near the few villages. | U |
| Sebuyau | 538 | 288 | 24 | 0 | 2.11 | Major town is Sebuyau (14,000 population), surrounded by a few villages. Other agricultural activities were observed. | U |
| Simunjan | 788 | 346 | 240 | 0 | 4.73 | Major town is Simunjan (22,000 population), a few villages. Two streams combine to form the main Simunjan River. One of the streams passes an oil palm mill which discharges into the river. | D |
| Sematan | 287 | 0 | 0 | 0 | 1.47 | Major town is Sematan (7,600 population), small villages. We observed agricultural activities by the local people. | U |
| Samusam | 163 | 0 | 0 | 0 | 0.85 | Well preserved tropical forest. Some peat in the upper catchment area. | U |

[1] Estimate is based on 'Wetlands International'. "Malaysia peat lands". Accessed through Global Forest Watch on 22nd November 2018 (www.globalforestwatch.org).

[2] Estimate is based on 'Oil palm concessions'. Accessed through Global Forest Watch on 22nd November 2018 (www.globalforestwatch.org).

[3] Estimate is based on 'Managed forest concessions'. Accessed through Global Forest Watch on 22nd November 2018 (www.globalforestwatch.org).

[4] Area estimates are based on the length and width of the primary course and main tributaries of the rivers. Length and width of the rivers were estimated using Google Earth (multiple readings).

[5] Estimate from (Staub et al., 2000).






Table 3: Overview of $N_2O$ and $CH_4$ concentrations, saturations and flux densities in rivers and estuaries of NW Borneo.

| River | Date | $N_2O$ | | | $CH_4$ | | |
|---|---|---|---|---|---|---|---|
| | | concentration nmol $L^{-1}$ | saturation % | flux density nmol $m^{-2}$ $s^{-1}$ | concentration nmol $L^{-1}$ | saturation % | flux density nmol $m^{-2}$ $s^{-1}$ |
| Rajang | Aug '16 | 2.0 – 14.1 | 28 – 215 | -0.33 – 0.48 | 13.2 – 233 | 719 - 9988 | 0.77 – 15 |
| | Mar '17 | 5.9 – 24.0 | 100 – 329 | 0 – 1.08 | 11.1 – 1008 | 455 – 40,598 | 0.34 – 62 |
| | Sept '17 | 18.6 – 24.6 | 277 – 390 | 0.76 – 1.22 | 7.4 – 150 | 350 – 6019 | 0.35 – 9.05 |
| Maludam | Mar '17 | 4.5 – 6.7 | 62 – 106 | -0.20 – 0.03 | 312 – 829 | 12,603 – 32,988 | 19 – 50 |
| | Sept '17 | 10.8 – 20.7 | 150 – 331 | 0.23 – 1.00 | 3.3 – 18 | 163 – 717 | 0.09 – 0.93 |
| Sebuyau | Mar '17 | 3.5 – 7.7 | 55 – 118 | -0.18 – 0.08 | 8.4 – 1228 | 396 – 50,774 | 0.41 – 78 |
| | Sept '17 | 12.8 – 23.0 | 176 – 335 | 0.36 – 1.08 | 6.4 – 29 | 299 – 1285 | 0.28 – 1.79 |
| Simunjan | Mar '17 | 2.5 – 41.4 | 35 – 570 | -0.31 – 2.20 | 39 – 1372 (14,999)[1] | 1642 – 57,459 (624,070)[1] | 2.37 – 88 |
| | Sept '17 | 5.1 – 26.5 | 73 – 365 | -0.13 – 1.24 | 2.5 – 21 | 106 – 878 | 0.01 – 1.18 |
| Sematan | Mar '17 | 4.3 – 8.2 | 71 – 109 | -0.11 – 0.04 | 8.6 – 12 | 433 – 47,055 | 0.43 – 72 |
| Samusam | Mar '17 | 4.0 – 9.5 | 67 – 142 | -0.13 – 0.19 | 16.5 – 978 | 830 – 43,807 | 0.95 – 63 |

[1] This extreme value was not included in further computations.





Table 4: Overview of N$_2$O and CH$_4$ flux densities from aqueous and soils ecosystems in SE Asia. (na stands for not available/not measured.)

| Site | Location | N$_2$O flux density, nmol m$^{-2}$ s$^{-1}$ | | CH$_4$ flux density, nmol m$^{-2}$ s$^{-1}$ | | Measurement or sampling dates | Reference |
|---|---|---|---|---|---|---|---|
| | | Range | Mean[1] | Range | Mean[1] | | |
| **Aqueous systems** | | | | | | | |
| Rajang River/Estuary | Sarawak, NW Borneo | -0.33 – 1.22 | 0.53 | 0.34 – 62 | 5.52 | Aug. 2016; Mar.; Sept. 2017 | This study |
| Maludam River/Estuary | Sarawak, NW Borneo | -0.20 - 1.00 | 0.32 | 0.09 – 50 | 15.9 | March 2017; September 2017 | |
| Sebuyau River/Estuary | Sarawak, NW Borneo | -0.18 – 1.08 | 0.39 | 0.28 – 78 | 15.4 | March 2017; September 2017 | |
| Simunjan River/Estuary | Sarawak, NW Borneo | -0.31 – 2.20 | 0.50 | 0.01 – 88 | 18.7 | March 2017; September 2017 | |
| Sematan River/Estuary | Sarawak, NW Borneo | -0.11 – 0.04 | -0.05 | 0.43 – 72 | 21.1 | March 2017 | |
| Samusam River/Estuary | Sarawak, NW Borneo | -0.13 – 0.19 | 0.05 | 0.95 – 63 | 21.7 | March 2017 | |
| Lupar River/Estuary | Sarawak, NW Borneo | 0.04 – 0.04 | *0.04* | 0.59 – 0.84 | *0.72* | June 2013; March 2014 | (Müller et al., 2016a) |
| Saribas River/Estuary | Sarawak, NW Borneo | 0.04 – 0.08 | *0.06* | 0.45 – 1.01 | *0.73* | June 2013; March 2014 | |
| Saribas River tributary | Sarawak, NW Borneo | 0.37 – 0.39 | *0.38* | 0.81 – 4.84 | *2.83* | June 2013; March 2014 | |
| Drainage canal, Kalimantan, settled | Central Kalimantan, S Borneo | -0.02 – 0.03 | 0 | 0 – 943 | 119 | September 2007; April 2008 | (Jauhiainen and Silvennoinen, 2012) |
| Drainage canal, Kampar, settled | Riau, eastern central Sumatra | 0.03 – 5.80 | 0.73 | 0 – 3672 | 776 | September 2007; April 2008 | |
| Drainage canal, Kampar, disturbed | Riau, eastern central Sumatra | 0.02 – 0.84 | 0.20 | 2.17 – 281 | 64.4 | September 2007; April 2008 | |
| **Soil systems** | | | | | | | |
| Forest | Sarawak, NW Borneo | -0.03 – 0.20 | *0.08* | -0.10 – 0.19 | *0.04* | August 2002 - July 2003 | (Melling et al., 2005, 2007) |
| Sago plantation | Sarawak, NW Borneo | 0.01 – 1.75 | *0.88* | -0.17 – 2.36 | *1.10* | August 2002 - July 2003 | |
| Oil palm plantation | Sarawak, NW Borneo | 0.01 – 0.58 | *0.29* | -0.76 – 0.11 | *-0.33* | August 2002 - July 2003 | |
| Undrained forest | Central Kalimantan, S Borneo | -0.09 – 1.16 | 0.02 | na | na | Dry/wet seasons in 2000/2001 | (Jauhiainen et al., 2012) |
| Drained forest | Central Kalimantan, S Borneo | -0.42 – 22.9 | 1.11 | na | na | Dry/wet seasons in 2001/2002; monitoring 2004 – 2007 | |
| Drained recovering forest | Central Kalimantan, S Borneo | -0.06 – 0.45 | 0.02 | na | na | Dry/wet seasons in 2001/2002; monitoring 2004 – 2007 | |
| Drained burned peat | Central Kalimantan, S Borneo | -0.70 – 0.88 | 0.11 | na | na | Dry/wet seasons in 2001/2002; monitoring 2004 – 2007 | |
| Agricultural peat in Kalampagan | Central Kalimantan, S Borneo | -0.95 – 0.89 | 0.12 | na | na | Dry/wet seasons in 2001/2002 | |
| Agricultural peat in Marang | Central Kalimantan, S Borneo | -0.86 – 0.59 | 0.07 | na | na | Dry/wet seasons in 2001/2002 | |
| Canopy soil of oil palm | Jambi, eastern central Sumatra | na | 0.001 | na | 0.0004 | February 2013 - May 2014 | (Allen et al., 2018) |
| Drained agricultural land (fertilized) | Various locations in SE Asia | 0.81 – 29.3 | 10.3 | 0.05 – 6.74 | 3.39 | Various dates | (Couwenberg et al., 2010): Review of results from various studies. |
| Drained, open vegetation (abandoned, not fertilized) | Various locations in SE Asia | -0.12 – 0.45 | 0.08 | na | na | Various dates | |
| Forested (drained and undrained peat swamp, agro-forestry) | Various locations in SE Asia | -0.06 – 1.51 | 0.39 | -0.73 – 11.6 | 5.45 | Various dates | |
| Rice paddies | Various locations in SE Asia | -0.04 – 0.23 | 0.07 | 7.17 – 98.1 | 52.7 | Various dates | |
| Peat soil | Various locations in SE Asia | na | na | 0 – 52.1 | 26.0 | Various dates | |
| Maludam Natl. Park | Sarawak, NW Borneo | na | na | na | 23.1 | November – December 2013 | (Tang et al., 2018) |

[1] Values in italics indicate a mean flux density computed from the range given in the table (when no mean flux density was given in the ref.)



Table 5: Mean annual emissions of $N_2O$ and $CH_4$ from rivers and estuaries in NW Borneo. The data
from Lupar and Saribas Rivers are from (Müller et al., 2016a).

| River | Emissions | |
|---|---|---|
| | Gg $N_2O$ yr$^{-1}$ | Gg $CH_4$ yr$^{-1}$ |
| Rajang | 0.33 | 1.27 |
| Maludam | 0.20 | 3.65 |
| Sebuyau | 0.24 | 3.53 |
| Simunjan | 0.32 | 4.30 |
| Sematan | -0.03 | 5.99 |
| Samusam | 0.03 | 4.99 |
| Lupar | 0.01 | 0.08 |
| Saribas | 0.01 | 0.04 |
| Sum | 1.09 | 23.8 |





Figure Captions

Figure 1: Map of the study area with locations of the sampling stations. Sampling stations from
August 2016 are displayed in red circles, from March 2017 in blue triangles, and from September
2017 in green diamonds. Major cities are highlighted in bold plus symbols. Inset is adapted from
(Staub et al., 2000).

Figure 2: $N_2O$ saturations along the salinity gradients of (a) Rajang, (b) Maludam, (c) Sebuyau, (d)
Simutan, (d) Sematan and (e) Samusam. The dashed lines indicate the equilibrium (100%) saturation.
The open cycles depict measurements from August 2016, the filled red cylces depict measurements
from March 2017 and the filled blue cycles depict measurements from September 2017.

Figure 3: $CH_4$ saturations along the salinity gradients of (a) Rajang, (b) Maludam, (c) Sebuyau, (d)
Simutan, (d) Sematan and (e) Samusam. The dashed lines indicate the equilibrium (100%) saturation.
The open cycles depict measurements from August 2016, the filled red cycles depict measurements
from March 2017 and the filled blue cycles depict measurements from September 2017.

Figure 4: (a) Mean $N_2O$ and (b) mean $CH_4$ concentrations for the individual rivers vs. the mean
monthly rainfall amount during the month of the sampling. We also included the mean $N_2O$ and $CH_4$
concentration for the Lupar, Saribas Rivers and Saribas tributary from (Müller et al., 2016a). The
linear correlation in (a) is described by $y = 0.08x + 5.76$ ($r = 0.72$, $n = 17$, significant at the 99% level).
The linear correlation in (b) is described by $y = -9.57x + 713.15$ ($r = 0.88$, $n = 13$, significant at the
99% level; please note that the encircled data were not included in the correlation).



6    Figures

Figure 1.

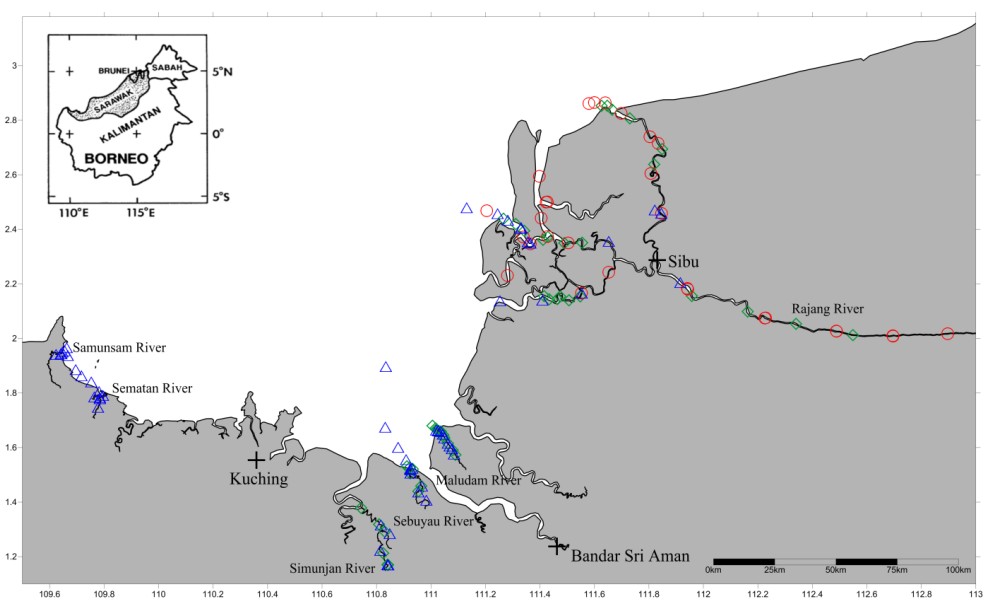





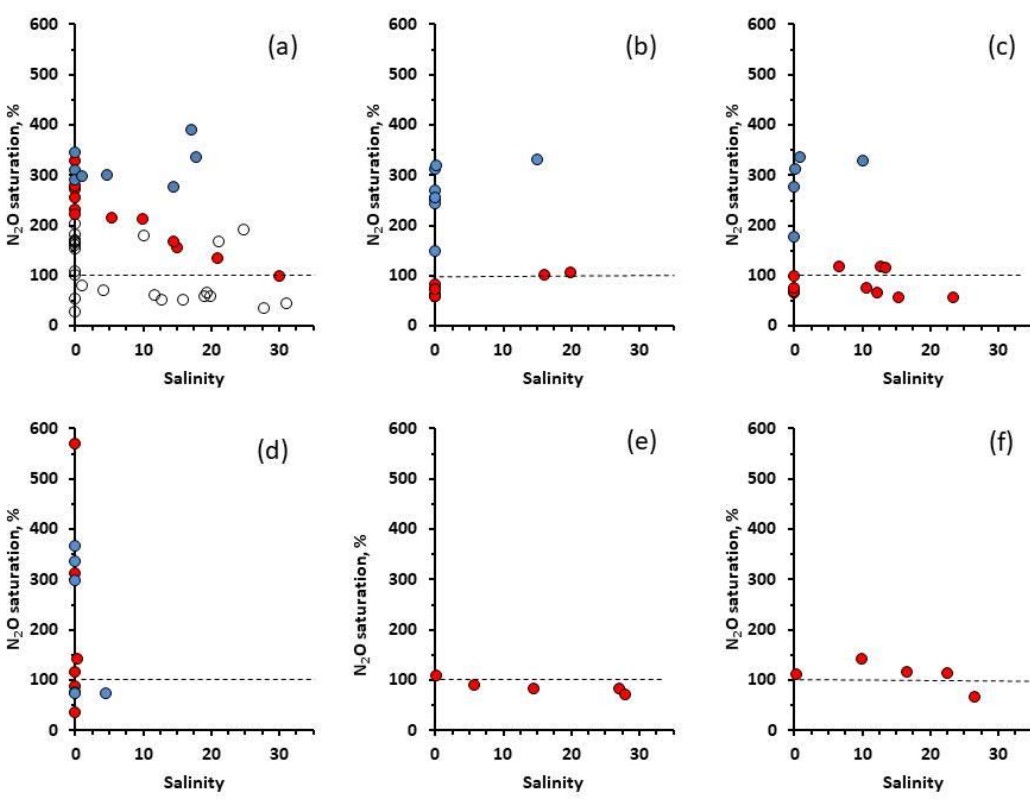

Figure 2.



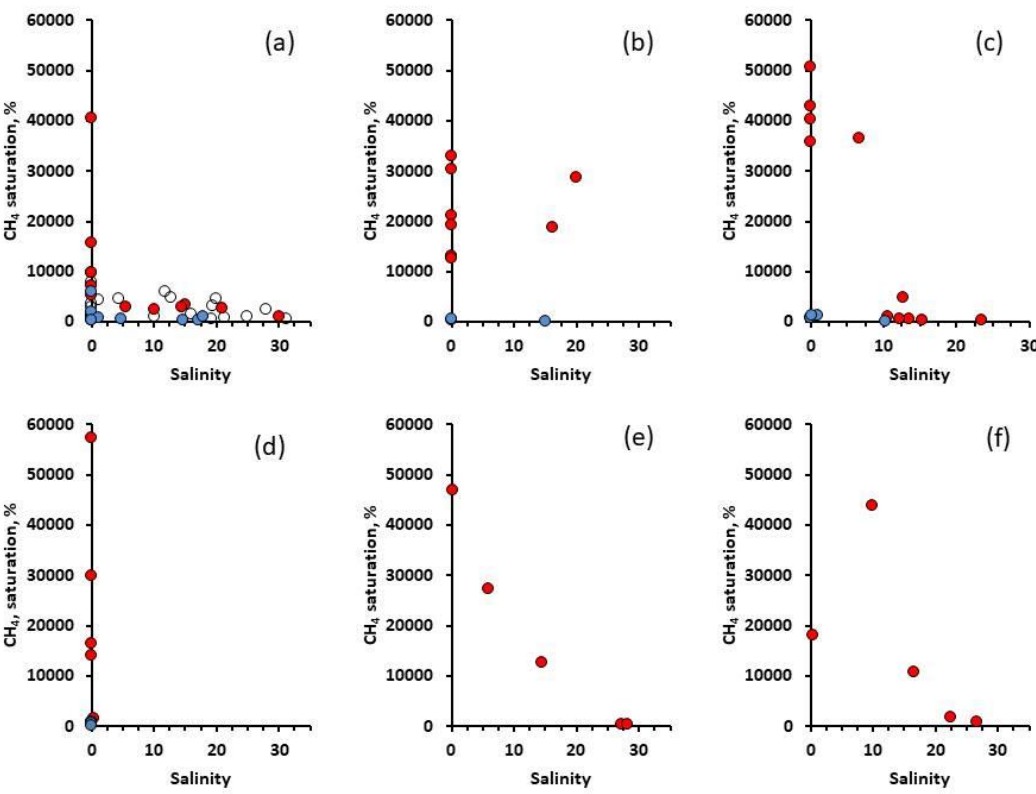

Figure 3.



Figure 4

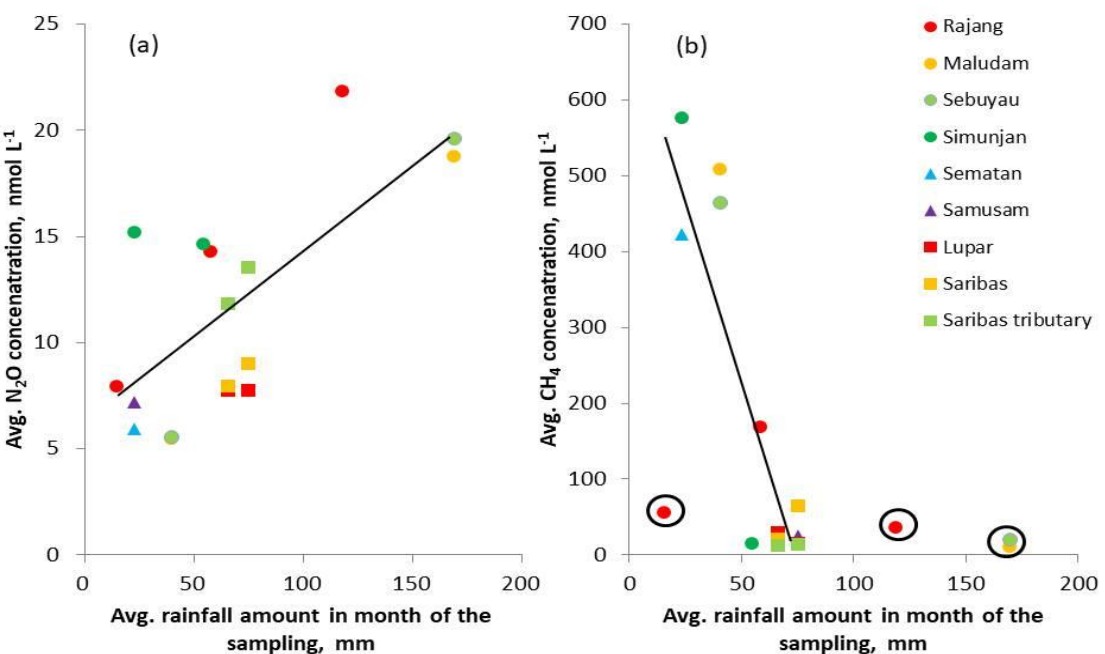