# Peer review of "Nitrous oxide $(N_2O)$ and methane $(CH_4)$ in rivers and estuaries of northwestern Borneo"

_Biogeosciences, 2019_

## Referee Comment (RC1) · Anonymous Referee #1 · 26 Jun 2019

The authors report a very valuable data-set of dissolved CH4 and N2O concentrations obtained in several estuaries in Borneo. I have a few minor suggestions for improvement/clarification listed below.

L 39 : Please provide ranges of pH, O2 and DOC. "very high/low" is vague.

L 77 : I suggest replacing "release" by "exchange", since the direction of the flux is not necessarily always to the atmosphere, as shown here by frequent N2O depletion in some rivers.

L 96 : CH4 is also oxidized aerobically in freshwater sediments, in rivers (Kelley et al. 1995) and lakes (Frenzel et al. 1990).

L 111: The number of references seems excessive to back a simple statement on the

occurrence of black water rivers in SE Asia.

L 142: Please specify how was the water collected for the CH4/N2O samples? Niskin bottle?

L152: Please provide the values of standards for N2O/CH4. Authors state that their standards were calibrated against NOAA standards, but NOAA standards have usually very low CH4/N2O values (close to atmospheric equilibrium), but given the reported concentrations, the measured pCH4 and pN2O should have strongly deviated from atmospheric equilibrium, unless the gas samples were diluted (in which case this needs to be specified).

L 166: Please specify how was pH measured.

L 171: Did you check if there was an interference of HgCl2 on NH4+ samples ? Based on personal experience HgCl2 strongly modifies NH4+ samples for colorimetric measurements.

L 247: Over-saturation of N2O of 12,480% was reported in an agriculture impacted small stream of the Meuse Basin (Borges et al. 2018).

L 256-273: The authors develop the idea that N2O production did not occur in black water rivers due to low pH values because of the protonation of NH3 and the pH-dependent reduction of nitrification and denitrification. Consequently, the authors conclude N2O production occurred in soils, and that N2O was subsequently transferred to the river. However, peat soils themselves are also very acid, so the same reasoning of inhibition of N2O production should also apply to soils. So, why should low pH inhibit N2O production in river water but not in soils?

L 256-273: The experiments of Le et al. (2019) showed that nitrification was strongly inhibited but still occurred until pH 5.3, and was totally inhibited at pH 5.0. Since N2O is produced as a by-product of nitrification, it is possible that the N2O yield increases with decreasing pH (the same way that N2O yield from nitrification increases with de-

creasing O2)? Even if this is not the case, the fact that nitrification is inhibited by pH but still occurs down to pH 5.3 still allows the possibility of N2O production occurring in river water in the sampled sites. So there could still be a case for N2O being produced in black-water rivers.

L 256-273: While the lowest values of pH in the ranges reported in Table 1 are clearly lower than 5.0 (the value at which nitrification was undetectable in the experiments of Le et al. (2019)), I cannot figure out how many observations of high N2O coincided with pH<5.0. It might be useful for the discussion to plot N2O versus pH to show the reader how many data point of high N2O occur at pH > and < 5.0.

L 315: Higher discharge/rain also leads to enhanced gas transfer velocities and loss of CH4 to the atmosphere. Higher discharge/rain also leads to decreased residence time of water (flushing of water), which will decrease the accumulation of CH4 in the water (even if sources such as sediment flux remain the same). Higher rain (surface runoff) also leads to simple dilution of all solutes (including CH4).

L 316: A negative relation between CH4 and discharge is not necessarily a general rule. Teodoru et al. (2015) reported higher CH4 in the Zambezi River during high-waters and lower CH4 during low-waters due to variable connectivity with floodplains. At a fixed station in the upper Congo, Borges et al. (2019) showed that the CH4 seasonal evolution roughly follows the one of discharge. So in both studies a positive relation between CH4 and discharge was reported.

L 318: Most of the low- vs high-water comparisons of MOX and CH4 given by Sawakuchi et al. are for white water and clear water rivers, and only for one black river at a single station (Negro). I'm not sure this is sufficient to derive a general rule on methane oxidation in black water rivers.

Further, methane oxidation is a first order process, so should be lower when CH4 concentrations are lower, so, it's unlikely that CH4 oxidation is higher when CH4 concentrations are lower, as stated by the authors.

L 363: I suggest replacing "results" by resulted

Please explain how were the "average" flux calculated. It's unclear how the "average" flux intensities and integrated fluxes were derived to take into account the estuarine geometry. Estuaries are generally wider at the mouth than upstream ("funnel shaped"). So even if high salinity regions show lower flux intensities, their relative contribution to total flux will have more weight (relative larger surface area). To put it in other words a simple average of all of the data points will lead to an over-estimation of the flux intensities because the average will be biased towards low salinity values that in reality correspond to a lower surface of estuary. So, each data point needs to be weighted by a corresponding surface area (section of the estuary), and the average should be surface weighted. This requires a little bit of GIS but is feasible (even with Google Earth).

Figures 2 & 3: it could be useful to add in plots a legend of the symbols.

References

Borges, A., Darchambeau, F., Lambert, T., Bouillon, S., Morana, C., Brouyère, S., Hakoun, V., Jurado Elices, A., Tseng, H.-C., Descy, J.-P., & Roland, F. (2018). Effects of agricultural land use on fluvial carbon dioxide, methane and nitrous oxide concentrations in a large European river, the Meuse (Belgium). Science of the Total Environment, 610–611, 342 - 355

Borges AV, F Darchambeau, T Lambert, C Morana, G Allen, E Tambwe, A Toengaho Sembaito, T Mambo, J Nlandu Wabakhangazi, J-P Descy, CR Teodoru, S Bouillon (2019) Variations of dissolved greenhouse gases ($CO_2$, $CH_4$, $N_2O$) in the Congo River network overwhelmingly driven by fluvial-wetland connectivity, Biogeosciences Discussions, doi: 10.5194/bg-2019-68

Frenzel P, Thebrath B, Conrad R (1990) Oxidation of methane in the oxic surface layer of a deep lake sediment (Lake Constance). FEMS Microbiol Ecol 73:149–158

Kelley CA, Martens CS, Ussler W III (1995) Methane dynamics across a tidally flooded riverbank margin. Limnol Oceanogr 40:1112–1129

Teodoru, C. R., Nyoni, F. C., Borges, A.V., Darchambeau, F., Nyambe, I., & Bouillon, S. (2015). Dynamics of greenhouse gases (CO2, CH4, N2O) along the Zambezi River and major tributaries, and their importance in the riverine carbon budget. Biogeosciences, 12(8), 2431-2453.

---

## Referee Comment (RC2) · Anonymous Referee #2 · 3 Jul 2019

Interactive comment on "Nitrous oxide (N2O) and methane (CH4) in rivers and estuaries of northwestern Borneo", by H.W. Bange et al

The available data set for greenhouse gas concentrations in tropical rivers/estuaries and the resulting emissions to air is small, so this paper makes a potentially valuable contribution in this area. The dataset was interesting and while some trends in the data were reflected in other studies cited, yet other studies found contrasting results that I felt were not duly considered or were ignored. I was therefore looking for some further discussion and my overall impression was that the treatment was a little simplistic in several areas. I therefore consider that some significant modifications to the text are required. These, and some additional minor comments, are listed below. L150: What certified gas standard values were used? L160-164: the mean relative errors of the

gas analyses were acknowledged as being rather high and this was ascribed to long storage times. What were the storage times and were these the same for all samples? If not, is there any statistically significant difference between the errors for samples stored for "long" vs "short" times? Also, was the greater sensitivity of $CH_4$ to storage shown by Wilson et al (2018) also the case here? This was not clear. L168-169: Was the pre-washing protocol described here and the method for collecting ancillary samples also used for gases? The description of dissolved gas sample collection was lacking in detail. L173: What was the precision of the DOC analyses? Supplementary data from another paper are cited but the precision should be stated here. Also, there does not seem to be any description of the method used for pH. L200: It would be useful to briefly consider the scale of the potential errors in the values of k600 applied. it was stated that mean values from another (seasonal) study were used but what was the range of values estimated in that study? These values were derived using rivers other than those studied here but are they morphologically similar?. As gas exchange in rivers is determined by river flow rates, depth, gradient and bedform, it would be worth commenting on whether these variables are similar for the study rivers and for those from which k600 was derived. L205: It was stated that the value of k600 used here was close to the mean value used in Alin et al (2011) but only their range, which is quite wide, was given. L210: Was monthly rainfall data the best resolution available and if not, why was it chosen? The rainfall data on the cited website seem to be available for hourly intervals so it would at least be useful to briefly consider the overall ranges for the months in question based on these higher resolution data. L236, Reference to Figure 2a. There is a spread of $N_2O$ (also $CH_4$) for some rivers at zero salinity, but given the resolution of these plots are these values all truly riverine or does the plot mask large changes taking place at very low salinities? It is important to unequivocally make this point. To show this more clearly it might be worth considering using composite plots in which the x-axis left of zero salinity is plotted as "distance upstream". That would clearly show the variability along the length of the catchment sampled and may help reveal any tributaries with different $CH_4/N_2O$ signatures from the main river

in each case. It was also stated that the decreasing trend of N2O with salinity was only linear in the Rajang in March, but given the errors inherent in the analyses couldn't the Simutan and Sematan (incidentally, these are both labelled "(d)" in the figure caption) also be linear? L256: The lack of overall trends for N2O (also CH4) with oxygen and nutrients are stated to be in-line with the results of Borges et al (2015) and Müller et al (2016a) but this is perhaps a bit dismissive of contrasting observations made in other studies. Richey et al (1998), Bouillon et al (2009), Borges et al (2015), Teodoru et al (2015) and Upstill-Goddard et al (2017), among others, did find clear correlations of N2O with oxygen and nutrients, and Upstill-Goddard et al (2017) noted that N2O vs oxygen could be positive or negative depending on river "type". Consequently, some wider discussion of the current findings within this context seems warranted. Line 280: Presumably the very high CH4 sample that was excluded from the discussion was real, and not an artefact. It would be worth stating this, unless there is some reason to suspect otherwise. Line 296-299: Could the explanation of decreasing CH4 with salinity be a little simplistic? At least one plot (figure 3f) would look almost conservative if the high value at around salinity 10 was excluded. Is the plot therefore indicating "removal" of CH4 between an intermediate estuarine "endmember" at salinity 10 and the seawater endmember? If so it would be instructive to estimate the degree of removal of the CH4 signal (by extrapolating the linear portion of the plot at high salinity back to zero and taking the ratio of that number to the salinity 10 value) that could then be ascribed to oxidation and/or gas exchange (notwithstanding that there are a very small number of data points in the plot). L304 (Section 3.4): I wonder how meaningful it is to plot mean N2O vs mean monthly rainfall. At the very least, some discussion of the likely errors in this approach might be necessary to establish its validity. Some questions are: is the relationship between rainfall and N2O constant over different timescales? is it always linear? Could there be a variable lag time following initial rainfall (the length of which might relate to rain intensity and duration and the duration of any dry periods between successive rain events) before the N2O signal appears in the rivers? What is the likely effect of rainfall on gas exchange (could suppress or enhance it) and simple dilution

(which relates to rainfall intensity). The relationship between rainfall, local hydrogeology and river flow may be complex and affect N2O processing in groundwater flow etc., so some more detailed discussion of the relationships between N2O and rainfall seems warranted. L315 onward: Upstill-Goddard et al (2017) found both positive and negative relationships between CH4 and oxygen in tropical rivers (Congo Basin) dependent upon river "type" (as for N2O), which was ascribed to the possible presence or absence of macrophytes (as also discussed earlier by Borges et al). The current results should be contrasted with these and other earlier findings. L345: It would be instructive to acknowledge the high degree of uncertainty in the flux estimates and to have some brief discussion of the likely major sources of these. Figure 2 and 3 captions. "cycles" should perhaps be "circles"

Bouillon, S., Yambélé, A., Spencer, R. G. M., Gillikin, D. P., Hernes, P. J., Six, J., Merckx, R., and Borges, A. V.: Organic matter sources, fluxes and greenhouse gas exchange in the Oubangui river (Congo River basin), Biogeosciences, 9, 2045–2062, doi:10.5194/bg-9-2045-2012, 2012 Borges, A. V., Abril, G., Darchambeau, F., Teodoru, C. R., Deborde, J., Vidal, L. O., Lambert, T., and Bouillon, S.: Divergent biophysical controls of aquatic CO2 and CH in the World's two largest rivers, Sci. Rep., 5, 5614, doi:10.1038/srep15614, 2015 Richey, J. E., Devol, A., Wofsy, S., Victoria, R., and Riberio, M. N. G.: Biogenic gases and the oxidation and reduction of carbon in Amazon river and floodplain waters, Limnol. Oceanogr., 33, 551–561, 1988. Teodoru, C. R., Nyoni, F. C., Borges, A. V., Darchambeau, F., Nyambe, I., and Bouillon, S.: Dynamics of greenhouse gases (CO2, CH4, N2O) along the Zambezi River and major tributaries, and their importance in the riverine carbon budget, Biogeosciences, 12, 2431–2453, doi:10.5194/bg-12-2431-2015, 2015. Upstill-Goddard, R.C., Salter, M.E., Mann, P.J., Barnes, J., Poulsen, J., and Dinga, B., Fiske, G.J. and. Holmes, R.M.: The riverine source of CH4 and N2O from the Republic of Congo, western Congo Basin. Biogeosciences, 14, 2267–2281, doi:10.5194/bg-14-2267-2017, 2017

---

## Author Comment (AC1) · 4 Sep 2019

**We thank Ref#1 for the comments which helped to improve the manuscript significantly.**

Anonymous Referee #1

The authors report a very valuable data-set of dissolved CH4 and N2O concentrations obtained in several estuaries in Borneo. I have a few minor suggestions for improvement/clarification listed below.

L 39 : Please provide ranges of pH, O2 and DOC. "very high/low" is vague.

***Reply (R): The ranges were added.***

L 77 : I suggest replacing "release" by "exchange", since the direction of the flux is not necessarily always to the atmosphere, as shown here by frequent N2O depletion in some rivers.

***R: We agree: 'release' was replaced with 'exchange'.***

L 96 : CH4 is also oxidized aerobically in freshwater sediments, in rivers (Kelley et al. 1995) and lakes (Frenzel et al. 1990).

***R: We added the missing information about aerobic CH4 oxidation in river sediments. However, we do not see a need to refer to studies of lake ecosystems. Moreover, aerobic CH4 oxidation in the river water is already mentioned in the next sentence.***

L 111: The number of references seems excessive to back a simple statement on the occurrence of black water rivers in SE Asia.

***R: We agree. The number of references has been reduced to three: 'Alkhatib et al., 2007; Martin et al., 2018; Moore et al., 2011'.***

L 142: Please specify how was the water collected for the CH4/N2O samples? Niskin bottle?

***R: S****amples were collected at 1 m depth using a Niskin sampler. We added this information to the text.***

L152: Please provide the values of standards for N2O/CH4. Authors state that their standards were calibrated against NOAA standards, but NOAA standards have usually very low CH4/N2O values (close to atmospheric equilibrium), but given the reported concentrations, the measured pCH4 and pN2O should have strongly deviated from atmospheric equilibrium, unless the gas samples were diluted (in which case this needs to be specified).

***R: We added the range of mole fractions of the used standard gas mixtures. These standards have been calibrated against certified NOAA gas standards in the laboratory at the MPI for Biogeochemistry in Jena, Germany. Unfortunately, the values of the primary gas standards are not known to us.***

L 166: Please specify how was pH measured.

***R: We added this information.***

L 171: Did you check if there was an interference of HgCl2 on NH4+ samples ? Based on personal experience HgCl2 strongly modifies NH4+ samples for colorimetric measurements.

*R: The indophenol blue method used here works well with low concentration of Hg. (We only added a tiny amount of HgCl2 solution (2-3 drops) into each bottle.) The precision of our method was frequently better than +/- 3%.*

L 247: Over-saturation of N2O of 12,480% was reported in an agriculture impacted small stream of the Meuse Basin (Borges et al. 2018).

*R: Thank you for pointing this out. We modified the text. We are now citing '(Borges et al., 2018)' instead of '(Barnes and Upstill-Goddard, 2011)'.*

L 256-273: The authors develop the idea that N2O production did not occur in black water rivers due to low pH values because of the protonation of NH3 and the pH-dependent reduction of nitrification and denitrification. Consequently, the authors conclude N2O production occurred in soils, and that N2O was subsequently transferred to the river. However, peat soils themselves are also very acid, so the same reasoning of inhibition of N2O production should also apply to soils. So, why should low pH inhibit N2O production in river water but not in soils?

*R: On the one hand, tropical soils indeed can have pH <4 and thus net N2O production should be low as well when adapting our line of arguments for rivers. On the other hand it is well known that significant N2O (peat) soil production occurs (mainly via denitrification) when the water table is high/the WFPS (water filled pore space) is 100% (Pihlatie et al., 2004; Regina et al., 1996). This is not necessarily a contradiction since the microbial community in tropical soils is probably very different to the one found in the rivers. Moreover, N2O production by denitrification seems to be generally less sensitive against low pH (see Blum et al. 2018).*

*Blum et al., The pH dependency of N-converting enzymatic processes, pathways and microbes: effect on net N2O production, Environmental Microbiology, 20, 1623-1640, 2018.*

*Pihlatie et al., Contribution of nitrification and denitrification to N2O production in peat, clay and loamy sand soils under different soil moisture conditions, Nutrient Cycling in Agroecosystems, 70, 135-141, 2004.*

*Regina et al., Fluxes of nitrous oxide from boreal peatlands as affected by peatland type, water table level and nitrification capacity, Biogeochemistry, 35, 401-418, 1996.*

L 256-273: The experiments of Le et al. (2019) showed that nitrification was strongly inhibited but still occurred until pH 5.3, and was totally inhibited at pH 5.0. Since N2O is produced as a by-product of nitrification, it is possible that the N2O yield increases with decreasing pH (the same way that N2O yield from nitrification increases with decreasing O2)? Even if this is not the case, the fact that nitrification is inhibited by pH but still occurs down to pH 5.3 still allows the possibility of N2O production occurring in river water in the sampled sites. So there could still be a case for N2O being produced in black-water rivers.

*R: Thank you for pointing this out. We agree and thus replaced 'unlikely' with 'low' which indeed better reflects a potential of N2O production in river waters at low pH.*

L 256-273: While the lowest values of pH in the ranges reported in Table 1 are clearly lower than 5.0 (the value at which nitrification was undetectable in the experiments of Le et al. (2019)), I cannot figure out how many observations of high N2O coincided with pH<5.0. It might be useful for the discussion to plot N2O versus pH to show the reader how many data point of high N2O occur at pH > and < 5.0.

*R: We added a new Figure 3 which shows N2O vs pH. We added 'Figure 3 shows the N2O concentrations along the pH gradients. Obviously there are no trends except for an enhancement of the N2O concentrations in September 2017.'*

L 315: Higher discharge/rain also leads to enhanced gas transfer velocities and loss of CH4 to the atmosphere. Higher discharge/rain also leads to decreased residence time of water (flushing of water), which will decrease the accumulation of CH4 in the water (even if sources such as sediment flux remain the same). Higher rain (surface runoff) also leads to simple dilution of all solutes (including CH4).

*R: Thank you for pointing this out. We modified the text which now reads: 'This relationship can be explained by an interplay of various processes such as: (i) decrease of CH4 concentrations caused by a higher water flow (i.e. dilution under the assumption that the net CH4 production does not change significantly), (ii) higher flux across the river/atmosphere interface during periods of higher discharge (caused by an enlarged river surface area and/or a more turbulent water flow) (Alin et al., 2011; Borges and Abril, 2011) and (iii) the enhancement of CH4 oxidation […].'*

L 316: A negative relation between CH4 and discharge is not necessarily a general rule. Teodoru et al. (2015) reported higher CH4 in the Zambezi River during high-waters and lower CH4 during low-waters due to variable connectivity with floodplains. At a fixed station in the upper Congo, Borges et al. (2019) showed that the CH4 seasonal evolution roughly follows the one of discharge. So in both studies a positive relation between CH4 and discharge was reported.

*R: We did not state that the negative CH4/discharge relationship is a general rule. Indeed we wrote '(i) the often observed inverse relationship […]' which clearly implies that it is not a general rule. So we do not see a need to revise the text at this point.*

L 318: Most of the low- vs high-water comparisons of MOX and CH4 given by Sawakuchi et al. are for white water and clear water rivers, and only for one black river at a single station (Negro). I'm not sure this is sufficient to derive a general rule on methane oxidation in black water rivers. Further, methane oxidation is a first order process, so should be lower when CH4 concentrations are lower, so, it's unlikely that CH4 oxidation is higher when CH4 concentrations are lower, as stated by the authors.

*R: Indeed, Sawakuchi et al. is cited erroneously by the reviewer: Sawakuchi et al. report MOx rates and CH4 isotopic signatures from four stations in two black rivers (three stations were located in the Rio Negro and one was located in the Rio Preto, see e.g. Tables 3 and 4 in Sawakuchi et al.). MOx rates were measured in the Rio Negro during both high and low water season and MOx was measured during high water in the Rio Preto. Moreover, Sawakuchi et al. concluded that 'the relative amount of MOx was maximal during high water in black and white water rivers and minimal in clear water rivers during low water'. Therefore, we have good reasons to follow the line of arguments by Sawakuchi et al. . (We agree, however, with the reviewer that more studies on this issue are needed.)*

L 363: I suggest replacing "results" by resulted

*R: We agree: 'results' was replaced with 'resulted'.*

Please explain how were the "average" flux calculated. It's unclear how the "average" flux intensities and integrated fluxes were derived to take into account the estuarine geometry. Estuaries are generally wider at the mouth than upstream ("funnel shaped"). So even if high salinity regions show lower flux intensities, their relative contribution to total flux will have more weight (relative larger surface area). To put it in other words a simple average of all of the data points will lead to an over-estimation of the flux intensities because the average will be biased towards low salinity values that in reality correspond to a lower surface of estuary. So, each data point needs to be weighted by a corresponding surface area (section of the estuary), and the average should be surface weighted. This requires a little bit of GIS but is feasible (even with Google Earth).

*R: We agree with the reviewer that a surface area-weighted estimate would give a more realistic 'picture' of the riverine emission estimates. However, since it was not possible to cover the entire salinity gradient during some of the sampling campaigns, an adequate surface area-weighted emission estimate is not possible for most of the rivers/estuaries sampled. Moreover, it seems reasonable to say that the uncertainty introduced by the poor seasonal/interannual coverage is much higher than the uncertainty introduced by the inadequate coverage of the salinity gradients (and thus the inadequate areal extrapolation).*

Figures 2 & 3: it could be useful to add in plots a legend of the symbols.

*R: The legends were added.*

---

## Author Comment (AC2) · 4 Sep 2019

**We thank Ref#2 for the comments which helped to improve the manuscript significantly.**

Anonymous Referee #2

The available data set for greenhouse gas concentrations in tropical rivers/estuaries and the resulting emissions to air is small, so this paper makes a potentially valuable contribution in this area. The dataset was interesting and while some trends in the data were reflected in other studies cited, yet other studies found contrasting results that I felt were not duly considered or were ignored. I was therefore looking for some further discussion and my overall impression was that the treatment was a little simplistic in several areas. I therefore consider that some significant modifications to the text are required. These, and some additional minor comments, are listed below.

L150: What certified gas standard values were used?

*Reply (R): The standard gas mixtures have been calibrated against certified NOAA gas standards in the laboratory of the MPI for Biogeochemistry in Jena, Germany. Unfortunately, the values of the primary gas standards are not known to us.*

L160-164: the mean relative errors of the gas analyses were acknowledged as being rather high and this was ascribed to long storage times. What were the storage times and were these the same for all samples? If not, is there any statistically significant difference between the errors for samples stored for "long" vs "short" times"? Also, was the greater sensitivity of CH4 to storage shown by Wilson et al (2018) also the case here? This was not clear.

*R: We added the mean storage time. Measurements of the Aug'16 samples were finished in Feb'17; measurements of the samples from the Mar'17 campaign were finished in Sept'17, and measurements of the samples from the Sept'17 campaign were finished in Feb'18. We did not see a trend of the mean relative error with storage time or a significant difference between the sampling campaigns.*

*We think that the greater sensitivity of CH4 samples is the reason for the higher mean relative error as described in Wilson et al. (2018). We modified the sentence which reads now 'The higher mean measurement error of the CH4 samples (compared to the N2O measurements) was attributed to the fact It was shown that CH4 samples are more sensitive to storage time than N2O samples (Wilson et al., 2018).'*

L168-169: Was the pre-washing protocol described here and the method for collecting ancillary samples also used for gases? The description of dissolved gas sample collection was lacking in detail.

*R: Water was collected at 1m depth using a Niskin sampler. Sample vials for N2O/CH4 were rinsed with sample water, filled to the maximum (without air bubbles), sealed on the spot using a crimper, and kept on ice for a maximum of 3 hours. When returned to the field station, HgCl2 was immediately added to stop any biological activity and samples were stored at 4 degree until shipment. We added the missing information to the text.*

L173: What was the precision of the DOC analyses? Supplementary data from another paper are cited but the precision should be stated here. Also, there does not seem to be any description of the method used for pH.

*R: We added the requested information on the precision of the DOC analyses and the method used for pH. DOC measurement performance was monitored using certified deep-sea water from the Hansell Laboratory, University of Miami (42–45 µmol L−1). Our analyses consistently yielded slightly higher values for the reference water, with a long-term mean (± 1 SD) of 47 ± 2.0 µmol L−1 (n = 51).*

L200: It would be useful to briefly consider the scale of the potential errors in the values of k600 applied. it was stated that mean values from another (seasonal) study were used but what was the range of values estimated in that study? These values were derived using rivers other than those studied here but are they morphologically similar?. As gas exchange in rivers is determined by river flow rates, depth, gradient and bedform, it would be worth commenting on whether these variables are similar for the study rivers and for those from which k600 was derived.

*R: We modified the text as requested:*

*1) The standard deviations of the k600 data given in Müller et al. (2016) were added.*

*2) We added 'Both rivers have very similar environmental and morphological settings in comparison to the rivers studied here.'*

*3) At the end of the section we added: 'kw in rivers depends on the turbulence at the river water/atmosphere interface, which in turn is mainly affected by water current velocity, water depth and river bed roughness and to a lesser extent by the wind speed (Alin et al., 2011; Borges and Abril, 2011). Since the k600 reported by (Müller et al., 2016a) were determined only during the wet season (March 2014), our mean k600 is biased because it does not account for a lower k600 which is to be expected during the dry season (resulting from a lower water current velocity (Alin et al., 2011)). This results in an overestimation of the flux densities.'*

L205: It was stated that the value of k600 used here was close to the mean value used in Alin et al (2011) but only their range, which is quite wide, was given.

*R: We added the requested information.*

L210: Was monthly rainfall data the best resolution available and if not, why was it chosen? The rainfall data on the cited website seem to be available for hourly intervals so it would at least be useful to briefly consider the overall ranges for the months in question based on these higher resolution data.

*R: The monthly rainfall data had been chosen because we think it is representative of the typical rainfall patterns. Indeed we refined our analysis by considering now the accumulated rainfall during up to four weeks prior to the date of sampling. For this we used rainfall data with a 3h resolution (available from the same website). We modified the text of Section 3.4 to account for this.*

L236, Reference to Figure 2a. There is a spread of N2O (also CH4) for some rivers at zero salinity, but given the resolution of these plots are these values all truly riverine or does the plot mask large changes taking place at very low salinities? It is important to unequivocally make this point. To show this more clearly it might be worth considering using composite plots in which the x-axis left of zero salinity is plotted as "distance upstream". That would clearly show the variability along the length of

the catchment sampled and may help reveal any tributaries with different CH4/N2O signatures from the main river in each case. It was also stated that the decreasing trend of N2O with salinity was only linear in the Rajang in March, but given the errors inherent in the analyses couldn't the Simutan and Sematan (incidentally, these are both labelled "(d)" in the figure caption) also be linear?

*R: In order to address the reviewers request we added a new Figure 3 which shows the N2O and CH4 concentrations along the pH gradients.*

*We do not think that the relationships for the Simunjan and Sematan Rivers are linear: Even when taking into account the associated measurement errors the data from the Simunjan River are well below a linear mixing line from endmembers at sal = 0 and sal = 30. There might be linear relationship for the Sematan River, but only when ignoring the data point at sal = 10.*

*We corrected the typos in the Figures captions.*

L256: The lack of overall trends for N2O (also CH4) with oxygen and nutrients are stated to be in-line with the results of Borges et al (2015) and Müller et al (2016a) but this is perhaps a bit dismissive of contrasting observations made in other studies. Richey et al (1998), Bouillon et al (2009), Borges et al (2015), Teodoru et al (2015) and Upstill-Goddard et al (2017), among others, did find clear correlations of N2O with oxygen and nutrients, and Upstill-Goddard et al (2017) noted that N2O vs oxygen could be positive or negative depending on river "type". Consequently, some wider discussion of the current findings within this context seems warranted.

*R: We added a sentence: 'There are, however, occasional observations in tropical rivers of N2O relationships with O2 and nutrients which were attributed to different river types such as swamp and savannah rivers (Upstill-Goddard et al., 2017).'*

*However, there does not seem to be a general (spatial or temporal) trend (we mentioned this in the introduction, see also Stanley et al., 2016). We think, therefore, that a more detailed discussion of results from other rivers (draining other ecosystems) does not improve our understanding of the results from peatland draining rivers presented here.*

*A some remarks about the references cited by the reviewer:*

*Bouillon et al. (2009) is missing in the reference list given by the reviewer. In the listed article (Bouillon et al., 2012) we could not find any N2O/O2 and N2O/nutrients correlations. There are no N2O data in Borges et al. (Sci Rep, 2015). The relationship of N2O with O2 mentioned in Richey et al., 1988, is far from being 'clear': 1) there are no statistics given and 2) the trend is only visible indirectly via plots of N2O/CO2 and AOU/CO2. The relationships of N2O with O2 or nutrients mentioned in Teodoru et al. (2015) are far from being 'clear': The authors state: 'There was no correlation between N2O and NH4+ or NO3-, while a positive relation with %DO was only found during wet seasons (data not shown).'*

Line 280: Presumably the very high CH4 sample that was excluded from the discussion was real, and not an artefact. It would be worth stating this, unless there is some reason to suspect otherwise.

*R: In fact we considered the very high CH4 concentration from the Simunjan River as real. In order to clary this point we replaced 'further computations' with 'emission estimates'.*

Line 296-299: Could the explanation of decreasing CH4 with salinity be a little simplistic? At least one plot (figure 3f) would look almost conservative if the high value at around salinity 10 was excluded. Is the plot therefore indicating "removal" of CH4 between an intermediate estuarine "endmember" at salinity 10 and the seawater endmember? If so it would be instructive to estimate the degree of removal of the CH4 signal (by extrapolating the linear portion of the plot at high salinity back to zero and taking the ratio of that number to the salinity 10 value) that could then be ascribed to oxidation and/or gas exchange (notwithstanding that there are a very small number of data points in the plot).

*R: A decrease of CH4 concentration with increasing salinity was observed in the majority of the measurements, see Fig 3 a,c,e and f. No trend was only observed for the data in Fig 3b. (in Fig 3d no measurements were available at salinities >0). Occasionally occurring higher CH4 concentrations were attributed to local point sources of CH4. So, we think that it is justified to state that there was a 'general decrease of CH4 with increasing salinity'.*

*We agree that the suggested idea is useful for estimating the riverine CH4 loss from the data presented in Fig 3f. However, we think that a (too) detailed interpretation of the data (based on only one river out of the six rivers measured) won't help to improve our general understanding of the CH4 trends in the rivers/estuaries of NW Borneo.*

L304 (Section 3.4): I wonder how meaningful it is to plot mean N2O vs mean monthly rainfall. At the very least, some discussion of the likely errors in this approach might be necessary to establish its validity. Some questions are: is the relationship between rainfall and N2O constant over different timescales? is it always linear? Could there be a variable lag time following initial rainfall (the length of which might relate to rain intensity and duration and the duration of any dry periods between successive rain events) before the N2O signal appears in the rivers? What is the likely effect of rainfall on gas exchange (could suppress or enhance it) and simple dilution (which relates to rainfall intensity). The relationship between rainfall, local hydrogeology and river flow may be complex and affect N2O processing in groundwater flow etc., so some more detailed discussion of the relationships between N2O and rainfall seems warranted.

*R: We refined our analysis by considering the relationship of the average N2O/CH4 concentrations with the accumulated rainfall from periods of up to four weeks prior to the date of sampling (= pre-sampling periods). (To this end, we now use rainfall data with a 3h resolution.) The linear N2O/rainfall relationship is quite robust and does not change when considering varying pre-sampling periods of accumulated rainfall prior to the dates of sampling. To address the question of a variable lag time we now consider periods of 1-4 weeks of accumulated rainfall prior to the dates of sampling. The resulting correlation coefficients are given in the new Table 6. Since the relationship of N2O and rainfall is robust over the given pre-sampling periods (1-4 weeks) we can conclude that the variability of time lag is negligible. (it would be great to have data on river discharge to answer this question, but these data were not available.) We modified this section.*

*The (revised) results for CH4 are more complex: Statistical significant linear relationships occur only when considering periods of 1 or 1.5 weeks before the dates of sampling. We modified this section as well.*

*To our knowledge the effect of rainfall on the trace gas exchange in rivers has not been investigated so far. We think, therefore, that a discussion about potential effects of rainfall (which is also highly variable in time and space) on riverine gas exchange is too speculative.*

L315 onward: Upstill-Goddard et al (2017) found both positive and negative relationships between CH4 and oxygen in tropical rivers (Congo Basin) dependent upon river "type" (as for N2O), which was ascribed to the possible presence or absence of macrophytes (as also discussed earlier by Borges et al). The current results should be contrasted with these and other earlier findings.

*R:* W*e added a sentence: 'There are, however, occasional observations in tropical rivers of CH4 relationships with O2 which were attributed to different river types such as swamp and savannah rivers (Upstill-Goddard et al., 2017).'*

*However, there does not seem to be a general (spatial or temporal) trend (we mentioned this in the introduction). We think, therefore, that a more detailed discussion of results from other rivers (draining other ecosystems) does not improve our understanding of the results of peatland draining rivers presented here.*

L345: It would be instructive to acknowledge the high degree of uncertainty in the flux estimates and to have some brief discussion of the likely major sources of these.

*R: We added '[…] (iii) the wind speed-driven gas exchange in estuaries is not adequately represented, and (iv) the mean k600 used here is most probably to high (see Section 3.3) resulting in an overestimation of the emissions.' However, we think that a detailed discussion about the inherent uncertainties of air/river exchange flux densities and emissions is beyond the scope of this article.*

Figure 2 and 3 captions. "cycles" should perhaps be "circles"

*R: We replaced 'cycles' with 'circles'.*